# Leptin Receptor Compound Heterozygosity in Humans and Animal Models

**DOI:** 10.3390/ijms22094475

**Published:** 2021-04-25

**Authors:** Claudia Berger, Nora Klöting

**Affiliations:** 1Medical Department III, Endocrinology, Nephrology, Rheumatology, CRC1052, University of Leipzig Medical Center, 04103 Leipzig, Germany; Claudia.berger@medizin.uni-leipzig.de; 2Helmholtz Institute for Metabolic, Obesity and Vascular Research (HI-MAG) of the Helmholtz Zentrum München at University of Leipzig, 04103 Leipzig, Germany

**Keywords:** leptin receptor, compound heterozygous mutation, mouse model

## Abstract

Leptin and its receptor are essential for regulating food intake, energy expenditure, glucose homeostasis and fertility. Mutations within leptin or the leptin receptor cause early-onset obesity and hyperphagia, as described in human and animal models. The effect of both heterozygous and homozygous variants is much more investigated than compound heterozygous ones. Recently, we discovered a spontaneous compound heterozygous mutation within the leptin receptor, resulting in a considerably more obese phenotype than described for the homozygous leptin receptor deficient mice. Accordingly, we focus on compound heterozygous mutations of the leptin receptor and their effects on health, as well as possible therapy options in human and animal models in this review.

## 1. Introduction

A compound heterozygote variant occurs if two or more different, recessive heterozygote mutations are present at the same locus on both alleles [1]. Compound heterozygous mutations are known to impact several diseases, such as the Brugada syndrome, infantile onset refractory epilepsy and restrictive dermopathy [2,3,4]. Pathogenic compound heterozygous variants of genes causing severe obesity are rarely reported, and most of them are located within the leptin receptor (LEPR) signaling cascade [5]. Thereby, compound heterozygous mutations are known in *LEPR*, proopiomelanocortin (*POMC*), melanocortin 4 receptor (*MC4R*), and proprotein convertase subtilisin/kexin type 1 (*PCSK1*), but not in leptin [5,6]. These mutations in the different genes have been described for fewer than 20 patients each [5]. Due to the low number of cases, the role of the environment and other possible genetic factors, a comparison between compound heterozygous and homozygous mutations is challenging. Here we review compound heterozygous mutations in human and animal models and gives an insight into the pathophysiology and therapy options.

## 2. Leptin Receptor

Leptin and LEPR signaling play a key role in regulating appetite, food intake, energy expenditure, glucose homeostasis, and fertility [7,8]. Leptin, also known as obese (*ob*) gene, was discovered as the molecule that causes a mutated form of severe obesity in *ob/ob* mice [9]. Leptin is an adipokine and is mainly secreted from white adipose tissue into the circulation [9]. While serum leptin levels decrease by fasting and increase with feeding, they also correlate with body fat mass and hyperleptinemia and can lead to leptin resistance [10,11,12]. The transport of leptin through the blood–brain barrier is not dependent on LEPR or passively transported, but the mechanism is still unclear [13]. In humans, mutations of leptin with severe early onset of obesity and hyperphagia has been diagnosed in fewer than 100 patients worldwide [5,14,15].

Lepr was discovered in *db/db* mice as a leptin-binding and membrane-spanning receptor expressed in the hypothalamus [16,17]. Also, mutations within the *Lepr* are rare, and only 2–3% of patients with severe early onset of obesity find that homozygous variants are the cause [5].

### 2.1. Structure of the Leptin Receptor

The extracellular part of the receptor is clustered in five domains (Figure 1): an *N*-terminal domain (NTD), with an unknown function and which is not necessary for signaling [18], followed by cytokine receptor homology (CRH) 1, immunoglobulin-like (Ig), CRH2 and fibronectin type III (FNIII) domain. CRH2 acts as the main leptin-binding domain and CRH1 as enhancer, which diminishes signaling if deleted [18,19]. Immunoglobulin-like domain (Ig) as well as fibronectin type III domain (FNIII) are essential for leptin binding and activation of the signaling [20,21].

The intracellular part of the receptor consists of the box1 motif, essential for Janus kinase 2 (JAK2) binding and signaling, and the three phosphorylatable tyrosine residues: Y985, Y1077 and Y1138 [22,23].

### 2.2. Isoforms

In humans, six different splicing variants of the *LEPR*—named LEP-Ra to LEP-Rf—are known. While all of them have the same N-terminal, extracellular sequence, they differ in their C-terminal, transmembrane and intracellular region [21]. LEP-Rb is the longest and only isoform that is highly expressed in the nuclei of the hypothalamus and regulates food intake and energy expenditure in humans and mice [25]. It is also expressed in adipose tissue, testis and peripheral blood cells in mice [26]. LEP-Ra, -Rc, -Rd and -Rf each have unique C-termini, and their function is still not completely understood. Hileman et al. suggested a role in the transport of leptin through the blood-brain barrier for LEP-Ra and -Rc [21,27]. While LEP-Rf only exists in humans and not in mice, LEP-Ra to -Re occur in both species. LEP-Ra is expressed in both species in lung, kidney, heart, liver and choroid plexuses [26,28].

The short and soluble LEP-Re is generated by proteolytic cleavage in humans and is a product of splicing in mice, directly secreted into the blood [29]. LEP-Re blood levels display the amount of membrane-bound receptors, with high levels indicating energy deficiency and low concentrations indicating a positive energy balance, reflecting leptin sensitivity in humans [30].

### 2.3. Leptin Receptor Signaling

The functional signaling is characterized by leptin binding to its receptor within the hypothalamus at neurons of the arcuate nucleus [31]. When leptin binds to its receptor in a stoichiometry of 1:1, it dimerizes or oligomerizes with other leptin–leptin receptor complexes [32,33] and induces the activation and self-phosphorylation of JAK2, as well as the autophosphorylation of the tyrosine residues Y985, Y1077 and Y1138 [23,34,35] (Figure 1). JAK2 itself activates different signaling pathways: signal transducer and activator of transcription 3 (STAT3), phosphatidylinositol 3-OH kinase/protein kinase B (PI3K/AKT) and extracellular signal-regulated kinase (ERK) pathway [34,35,36,37]. The STAT3 pathway is activated by phosphorylation of Y1138, which leads to STAT3 recruitment and phosphorylation by JAK2, followed by pSTAT3 translocation into the nucleus. There it acts as a transcription factor of suppressor of cytokine signaling 3 (SOCS3) and POMC, and inhibits agouti-related protein (AGRP) [34,36,38,39,40]. SOCS3 acts as a feedback inhibitor of JAK2 [34], while activation of anorexic POMC and inhibition of orexic AGRP expression leads to reduced food intake [39,40]. Taken together, STAT3 regulates food intake and energy balance [41].

In the second pathway (PI3K/AKT), phosphorylated JAK2 phosphorylates insulin receptor substrate (IRS), resulting in PI3K and AKT activation. In turn, pAKT phosphorylates forkhead box protein O1 (FOXO1), which then translocates from the nucleus to the cytoplasm [37]. The translocation of FOXO1 stops the expression of orexic AGRP gene and increases the expression of the anorexic POMC gene, followed by reduced food intake [42,43]. Thereby, translated POMC is processed by PCSK1 and results in releasing α-melanocyte-stimulating hormone (α-MSH), which activates the melanocortin 4 receptor (MC4R), finally leading to satiety [44,45]. In the fasting state, AGRP binds to MC4R as a competitive α-MSH antagonist, causing increased hunger [44].

Third in the process, the ERK-pathway is activated. Recruited by phosphorylated Y985, Src homology region 2 domain-containing phosphatase-2 (SHP2) activates growth factor receptor-bound protein 2 (GRB2), followed by ERK activation [46]. Activated ERK is important for the regulation of energy homeostasis [34].

Taken together, LEPR signaling is a complex interactive network, where every part of the receptor is necessary for a functioning regulation of hunger and satiety.

### 2.4. Function

Besides hunger and satiety, the loss of LEPR function causes infertility, accelerated growth, disrupted pubertal development [47,48], metabolic disorders such as insulin resistance [49], impaired thyroid and immune function [50,51].

Women suffering from mutations in leptin or LEPR can be infertile due to low levels of follicle-stimulating and luteinizing hormones, leading to the loss of pubertal maturity and hypogonadotropic hypogonadism [48,52]. On the other hand, it is reported that men with leptin receptor mutations achieve fertility at the beginning of puberty [53]. Fertility depends on the location of the mutation, and while STAT3 and 5 signaling do not influence fertility, the exact mechanism is not yet known [41,54]. Also, growth is not dependent on STAT3 signaling of LEPR, as studies of STAT3 in Lepr-deficient mice exhibited increased growth, while whole-brain knockout animals demonstrated reduced body length [41,55]. Reports on growth in humans with impaired LEPR function are very heterogeneous. Decreased growth hormone response and insulin-like growth factor 1 (IGF1) concentrations or Leptin as counterpart are still discussed.

Leptin signaling is also able to regulate glucose homeostasis, as shown in *ob/ob* mice [56]. Thereby, the signaling is taking place in the hypothalamus, as well as within the PI3K part of the Lepr signaling pathway [41,57,58].

The nutritional status and concentrations of leptin within serum modulate the T-cell immune function [59]. Thereby, low leptin levels lead to a reduced CD4+ T-cell number, a decreased T-cell proliferation, and a higher compensatory B-cell number, leading to an altered cytokine release after nonspecific and specific stimuli [48,60].

### 2.5. Inheritance Pattern of Human LEPR Mutation and Functional Analysis

LEPR mutations are inherited in an autosomal–recessive pattern. As shown by Nunziata et al., most homozygous patients are born of consanguineous parents, and obesity due to LEPR mutations aggregates in cultures with consanguineous marriages [61,62]. In contrast, compound heterozygous mutations appear in progeny of non-consanguineous parents [5].

To determine if obesity is of monogenic origin, the DNA of subjects has to be sequenced and further analyzed—at least the genes of the LEPR signaling cascade. Using computational algorithms, it is now possible to predict structural and functional changes resulting from single mutations. As shown by Gandhi Muruganandhan and Manian, single nucleotide polymorphisms (SNPs) within the *LEPR* gene result in structural and functional changes like changing the binding affinity of ligand and receptor [63]. Other SNPs within the *LEPR* are associated with a higher BMI in a Korean or Spanish population [64,65], insulin resistance or adiponectin serum level [64], as well as obesity and leptin serum level [66]. Thereby, the analysis of SNPs helps to identify disease-causing variants in a gene within a population [67].

Moreover, Voigtmann et al. used RaptorX, a computational protein structure prediction technique, and further analyzed the receptor models with Rosetta to predict the impact of the different mutations [68]. Approaches like these will help to predict the pathogenicity of new mutations.

## 3. Pathogenesis of Leptin Receptor Mutations

In 1998, Clément et al. reported the first pathogenic *LEPR* mutation after assessing a consanguineous family of Kabyle origin, including three children with early-onset morbid obesity and short stature [52]. To the best of our knowledge, 40 different pathologic variants of the *LEPR* gene have been described in 52 published cases so far [19,47,48,53,61,68,69,70,71,72,73,74,75,76].

Until now, only eight patients were described with compound heterozygous mutations within the *LEPR* gene, and two of them were siblings with the same mutation (Table 1) [68,71,77,78,79]. All of them developed early-onset severe obesity from the first days of life, with pronounced hyperphagia reaching a BMI of 29.7 and 33.1 kg/m^2^ for the nine-year-old girls and 41.6–67.7 kg/m^2^ for the adult subjects [68,71,77,78,79]. Due to the possible defect in signaling and the extended adipose tissue mass, strongly elevated leptin levels were measured in three subjects [71]. In contrast, three others exhibited leptin levels comparable to other people with obesity [48,68,77]. In the two remaining cases, leptin levels were not reported [79]. Hypogonadotropic hypogonadism was also described as a symptom for two [79], while three patients did not suffer from it [71,77]. Another symptom described was the altered growth induced by impaired levels of growth hormone. Thereby, an increased [77] or decreased [48,68,79] growth was detected. Additionally, two subjects suffered from hyperinsulinemia as well as dyslipidemia [68,79]. Type 2 diabetes, hepatic steatosis, hyperuricemia and hypothyroidism were described for one patient each [68,71,79].

Several compound heterozygous *LEPR* mutations result in diverse effects on human health. Therefore, it can be considered likely that these mutations are widespread within the *LEPR* gene. However, while the range is between 1264 bp and 2227 bp at *LEPR* gene (protein: 422–743 aa), all of them are located within the extracellular part of the receptor (Figure 2). Hence, in six of seven variants, these mutations lead to a truncated or probably truncated protein with decreased or abolished activation of LEPR signaling. 

Altered LEPR signaling was investigated in two studies. Voigtmann et al. used Hek293 cells expressing either fluorescence-tagged wildtype LEPR proteins or the mutant mimicking the compound heterozygous phenotype. In cell culture, they analyzed the surface expression, leptin binding, and signal transduction. They pointed out that the compound heterozygous variant showed reduced incidence at the cell surface with high intracellular protein retention, reduced binding affinity, and a loss of pSTAT3 activation [68]. In the second study, Kimber et al. analyzed the Arg612His mutation with a transfected Hek293 cell model regarding cell surface expression, ligand binding and STAT3 activation. There, the homozygous mutated variant showed a reduced surface expression, a mostly reduced leptin binding, and a low residual pSTAT3 activity. Neither the second mutation of the compound heterozygous variant nor the mixed variant was analyzed in the cell model [78]. Both cell models gave an impression of the changed molecular mechanism causing hyperphagia and the severe obese phenotype.

## 4. Therapy Options

For a long time, bariatric surgery and/or restricted food intake were considered the only therapy options for patients suffering from severe obesity. Since November 2020, the MC4R agonist Setmelanotide was approved by the U.S. Food and Drug Administration (FDA) (see Table 1) [80,81]. The phase 3 clinical trial included 11 participants (6 with compound heterozygous and 5 with homozygous *LEPR* mutations) treated over 52 weeks. Setmelanotide was injected subcutaneously once daily while the dose was adjusted for every patient individually within the first weeks. After 12 weeks on therapeutic dose, 7 participants reached the 5 % weight-loss threshold and continued within the study. At the end of the trial period, 5 of the 11 participants lost at least 10 % of weight. Hunger was reduced for 8 of the 11 participants by at least 25 %, as measured by a questionnaire. During the treatment, several adverse events, like injection site reaction (11 participants), skin hyperpigmentation (4 participants), and nausea (4 participants) occurred. In addition, one participant developed grade 1 hypereosinophilia, possibly related to Setmelanotide, and had to end the trial.

Other therapy options are controlled and restrictive diets, physical activity programs, psychomotricity and hormone substitution, including the familial environment, to increase the efficacy of the treatment [5,71]. The success of a bariatric surgery depends on the method as well as the familial environment, while vertical gastroplasty seems to be the best variant for monogenetic obesity, according to Huvenne et al. [5].

Taken together, this is a milestone for people suffering from *LEPR* mutations, especially since a bariatric surgery does not cure the cause and does not show a good long-term outcome [19]. Setmelanotide should be considered as a therapy option while taking into account the side effects.

## 5. Animal Models of Compound Heterozygous *Lepr* Mutations

Interestingly, animal models of compound heterozygous *Lepr* mutations are even rarer than human cases, and there are no spontaneous cases on record so far. Only two experimentally designed mouse models have been published: one is from 1998 and describes a mixed model of *ob/ob* and *db/db* mice, which is actually not heterozygous on the same gene, but still has a heterozygous *Lepr* mutation; the other is our previously published C57BL/6N-*Lepr*^L536Hfs*6-1NKB/db^ (*Lepr*^L536Hfs*6/db^) model [82,83].

The *Lepr*^L536Hfs*6/db^ model develops an even more early-onset severe obese phenotype and body fat mass than both homozygous parental strains [82]. Both mutations originally occurred spontaneously, were backcrossed onto the same C57BL/6N background, and crossed by heterozygous breeding to create the new compound heterozygous mouse model [16,82]. The compound heterozygous model might be of particular interest to better understand how allelic and locus heterogeneity affect variation in human genotype–phenotype associations.

Additionally, in a study focused on the characterization of a homozygous *Lepr* mutation created by CRISPR/Cas9 system, a compound heterozygous rat strain is named but not characterized [84]. Furthermore, in canine and feline models, only homozygous mutations within the Lepr signaling cascade are described [85]. In conclusion, the only known, characterized animal model with compound heterozygous *Lepr* mutation is the *Lepr*^L536Hfs*6/db^ model.

## 6. Conclusions

Compound heterozygous mutations in *LEPR* are exceedingly rare. To guarantee the best treatment, subjects with early-onset severe obesity should be sequenced in forehand of a bariatric surgery, and preferably treated with the new therapeutic Setmelanotide if a mutation in *LEPR* or *POMC* is detected. For a better understanding of the effects of different mutations on developing obesity, transgenic cell or animal models are a good option to determine damaged signaling or residual activity.

## Figures and Tables

**Figure 1 ijms-22-04475-f001:**
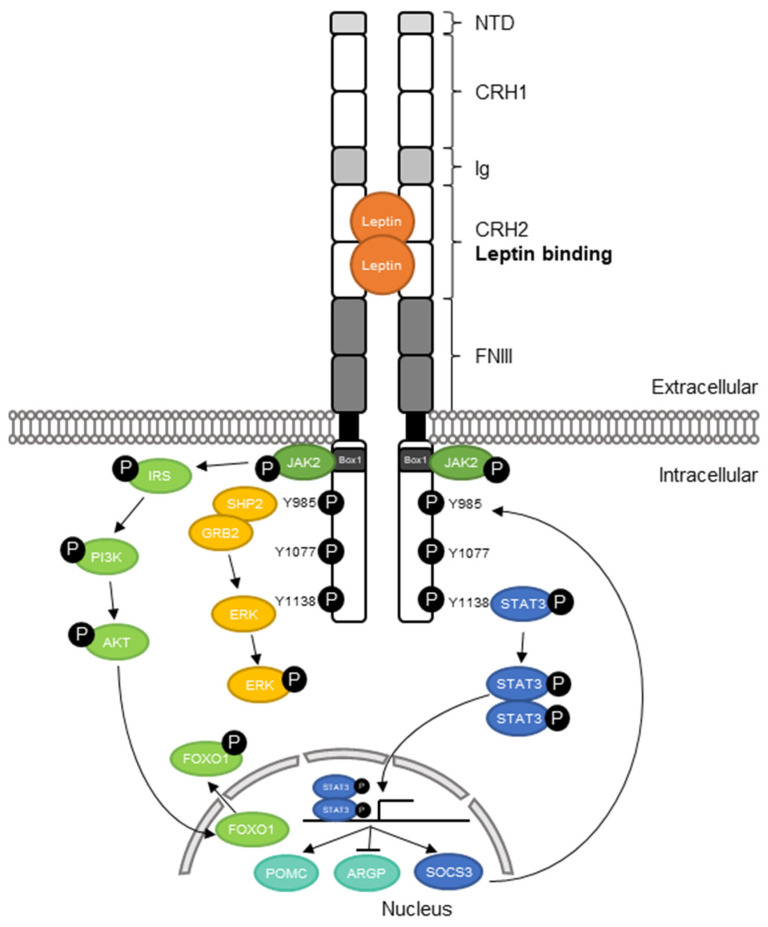
Leptin receptor structure and signaling. With the binding of leptin at its receptor, JAK2 is activated and the Y-residues phosphorylated, and STAT3 is recruited to Y1138 and phosphorylated. Activated pSTAT3 forms homodimers and translocates into the nucleus, where it promotes POMC expression and the inhibition of AGRP mediating a satiety signal. Phosphorylated JAK2 phosphorylates IRS, which activates the PI3K/AKT signaling pathway. In turn, pAKT phosphorylates FOXO1, resulting in translocation of pFOXO1 and a suppression of food intake. Phosphorylation of Y985 recruits and activates SHP2 and GRB2, which results in ERK phosphorylation, regulating energy homeostasis. NTD: *N*-terminal domain; CRH: cytokine receptor homology; Ig: immunoglobulin-like domain; FNIII: fibronectin type III; JAK2: Janus kinase 2; IRS: insulin receptor substrate; PI3K: phosphatidylinositol 3-OH kinase; AKT: serine-threonine kinase; SHP2: Src homology region 2 domain-containing phosphatase-2; GRB2: growth factor receptor-bound protein 2; ERK: extracellular signal-regulated kinase; STAT: signal transducer and activator of transcription; POMC: proopiomelanocortin; SOCS3: suppressor of cytokine signaling 3. Modified from Kwon et al. [24].

**Figure 2 ijms-22-04475-f002:**
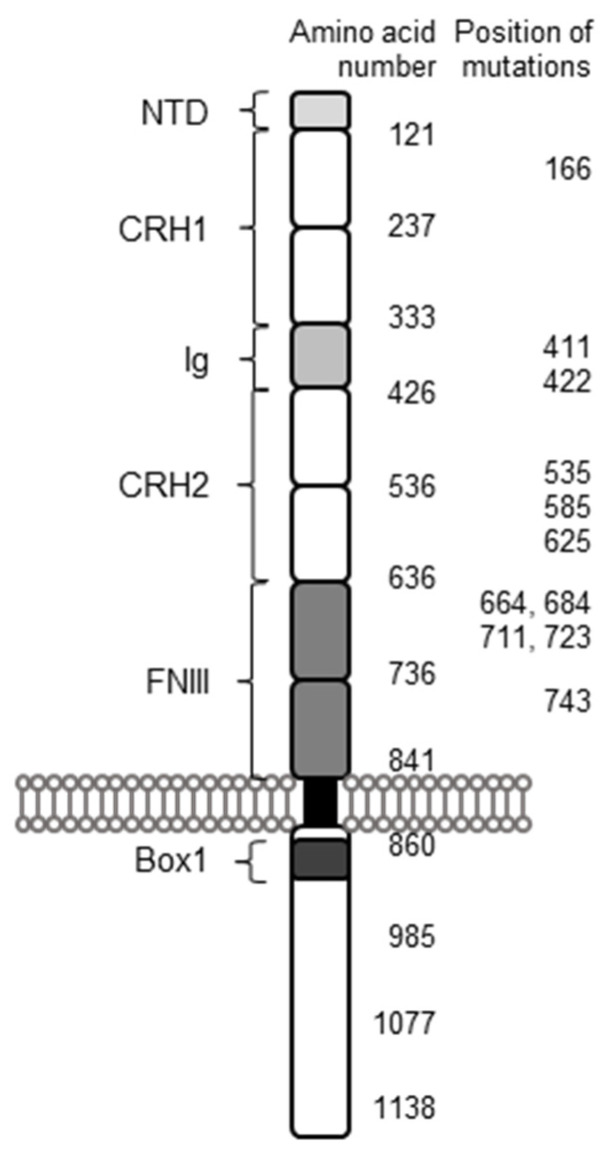
Model of a leptin receptor with different domains, their location and location of known mutations of compound heterozygous patients. All variants are located within the extracellular part of the receptor. NTD: *N*-terminal domain; CRH: cytokine receptor homology; Ig: immunoglobulin-like domain; FNIII: fibronectin type III domain. Adapted from Peelman et al. [21].

**Table 1 ijms-22-04475-t001:** List of different published LEPR mutations on gene and protein level with effects on LEPR itself, as well as on the health of the subjects and their treatment.

	Mutation Gene Level	Mutation Protein Level	Protein	Symptoms	**Treatment**
Kimber 2008, Farooqi 2007 [48,78]	1 bp deletion in codon 15 and Arg612His	- Arg612His	“Receptors with some residual ability to phosphorylate STAT3 in response to leptin”	-Early extreme obesity -Hyperphagia -Lep not elevated -Impaired growth	Not described
Huvenne 2015 [71]	c.1604–1G > A and c.Δexon6–8	p.535–1G>A (probably exon 12 skipping) and p.Pro166CysfsX7	Probably truncated protein and truncated protein (172 aa)	-Early extreme obesity -Hyperphagia -Elevated leptin level	Care models described
	c.1264T > C and c.2131 dup	p.Tyr422His and p.Thr711AsnfsX18	Probably damaged and truncated protein	-Early extreme obesity -Hyperphagia -Elevated Lep level -Type 2 diabetes (1 of sibling) -No Hypogonadotropic hypogonadism	Care models described
Hannema 2016 [77]	c.1753–1dupG c.2168C > T	p.Met585Aspfs * 2 p.Ser723Phe	Probably truncated protein	-Early extreme obesity -Normal Lep for obese -Increased growth -No Hypogonadotropic hypogonadism	Not described
Zorn 2020 [79]	c.2598-3_2607delTA- GAATGAAAAAG c.2227 T > C	Intron p.Ser743Pro	Truncated protein	-Early extreme obesity -Hyperphagia -Hypogonadotropic hypogonadism -Dyslipidemia -Hyperinsulinemia -Hepatic steatosis -Hyperuricemia	-Behavioral treatments -Bariatric surgery -MC4R agonist Setmelanotide therapy
	c.1874G > A c.2051A > C	p.His684Pro p.Trp625 *	Truncated protein	-Early extreme obesity -Hyperphagia -Hypogonadotropic hypogonadism -Growth hormone deficiency	-Conservative treatment -Gastric banding and removal -MC4R agonist Setmelanotide therapy
Voigtmann 2020 [68]	c.1231_1233 c.1835G > A	p.Tyr411del p.Trp664Arg	Probably truncated protein	-Early severe obesity -Hyperphagia -Short statue -Hypothyroidism -Dyslipidemia -Hyperinsulinemia -Normal Lep for obese	-MC4R agonist Setmelanotide in clinical trail

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
