# Peer review of "Leptin Receptor Compound Heterozygosity in Humans and Animal Models"

_ijms, 2021, doi:10.3390/ijms22094475_

Round 1

Reviewer 1 Report

Berger et al. present a mini-review of compound heterozygosis mutations of the leptin receptor, trying to focus in their effects on health and therapies available for this alterations. In general, the manuscript is well organised and sounds good. The different sections are small, but they content clear information, so for a mini-review I consider that it is fine. However, some points must be modified to make this manuscript suitable for publication.

Major comments:

-in the abstract, the authors claim that they are going to focus in the effects on health and possible therapies mainly; however, although the table 1 describes and organises really well symptoms and treatment, I consider that this should be discussed with more detail in the text, increasing the health problems associated in section 3 and extending a little the section 4.

-I think a small section talking about leptin (production, transport) could make the manuscript more complete, even being a mini-review; for example, between section 1 and 2.

-in mi opinion, section 2.1 and section 2.2 should be reorganised: first 2.1 Structure of leptin receptor; then 2.2 Isoforms.

-section 2.4: it is too small; the authors has to extend it.

-the text should contain references to the figures; just table 1 is referred.

Minor comments:

-the authors should make uniform e the way that they name leptin receptor. Sometimes it appears as LEPR (line 157, 171, 172, etc), others Lepr (line 51). Also, I consider that when they are talking about the gene, they should use Lepr using cursive.

-some references were not inserted, and this has to be corrected: line 68, line 173,

-Figure 1: this figure is breaking the continuity of the text on the figure legend, lines 81 and 82 should be moved to the bottom of the figure.

-Table 1: make uniform the beginning of the sentences to capital letter or not, but some of the symptoms and treatment start with capital letter and others no.

Author Response

Major Comments:

  • In the abstract, the authors claim that they are going to focus in the effects on health and possible therapies mainly; however, although the table 1 describes and organises really well symptoms and treatment, I consider that this should be discussed with more detail in the text, increasing the health problems associated in section 3 and extending a little the section 4.

We thank the reviewer for the positive feedback and followed the idea of extending section 3 and 4.

All of them developed early-onset severe obesity from the first days of life with pronounced hyperphagia reaching BMI of 29.7 and 33.1 kg/m2 for the 9-year-old girls and 41.6-67.7 kg/m2 for the adult subjects [65,66,74–76].

Other therapy options are controlled and restrictive diets, physical activity pro-grams, psychomotricity and hormone substitution including the familial environment to increase the efficacy of the treatment [71,5]. The success of a bariatric surgery de-pends on the method as well as the familial environment, while vertical gastroplasty seems to be the best variant for monogenetic obesity according to Huvenne et al. [5].

Setmelanotide should be considered as a therapy option taking into account the side effects.

  • I think a small section talking about leptin (production, transport) could make the manuscript more complete, even being a mini-review; for example, between section 1 and 2.

We are thankful for this comment and agree with the reviewer. We included a section about leptin within the manuscript:

Leptin is an adipokine and mainly secreted from white adipose tissue into the circulation [9]. While serum leptin levels decrease by fasting and increase with feeding, they also correlate with body fat mass and hyperleptinemia could lead to leptin resistance [10–12]. The transport of leptin through the blood brain barrier is not depending on the LEPR or passively transported, but the mechanism is still unclear [13].

  • In my opinion, section 2.1 and section 2.2 should be reorganised: first 2.1 Structure of leptin receptor; then 2.2 Isoforms

We thank the reviewer for the advice and reorganized the two parts.

  • section 2.4: it is too small; the authors has to extend it

We thank the reviewer for the feedback and extended the section 2.4 with the following part:

Women suffering from mutations in leptin or LEPR could be infertile regarding to low levels of follicle-stimulating and luteinizing hormone leading to missing attain of pubertal maturity and hypogonadotropic hypogonadism [52,48]. On the other hand, it is reported, that men with leptin receptor mutations getting fertile with the beginning of puberty [53]. Fertility depends on the location of the mutation, while STAT3 and 5 signaling do not influence fertility, the exact mechanism is not known jet [41,54]. Also, growth is not depending on STAT3 signaling of LEPR, as studies with STAT3 in LEPR deficient mice exhibited increased growth while whole brain knock out animals demonstrated reduced body length [41,55]. Reports on growth in humans with im-paired LEPR function are very heterogeneous. Decreased growth hormone response and insulin-like growth factor 1 (IGF1) concentrations or Leptin as counterpart are still discussed.

Leptin signaling is also able to regulate glucose homeostasis as shown in ob/ob mice [56]. Thereby, the signaling is taking place in hypothalamus as well as within the PI3K part of the LEPR signaling pathway [41,57,58].

The nutritional status and concentrations of leptin within serum modulate the T-cell immune function [59]. Thereby, low leptin levels lead to a reduced CD4+ T-cell number, a decreased T-cell proliferation and a higher compensatory B-cell number leading to an altered cytokine release after nonspecific and specific stimuli [60,48].

  • The text should contain references to the figures; just table 1 is referred.

We thank for this important comment. The references appeared with an error message due to technical problems with Microsoft Word document converting into a pdf file. We fixed this and the references are now visible.

Minor Comments:

  • The authors should make uniform e the way that they name leptin receptor. Sometimes it appears as LEPR (line 157, 171, 172, etc), others Lepr (line 51). Also, I consider that when they are talking about the gene, they should use Lepr using cursive.

We are sorry for the different spelling of the abbreviation and changed it into a uniform appearance for genes and proteins.

  • Some references were not inserted, and this has to be corrected: line 68, line 173,

We thank for this important comment. The references appeared with an error message due to technical problems with Microsoft Word document converting into a pdf file. We fixed this and the references are now visible.

  • Figure 1: this figure is breaking the continuity of the text on the figure legend, lines 81 and 82 should be moved to the bottom of the figure.

We thank for this hint. The break occurred after converting the manuscript into a pdf file. We corrected the problem.

  • Table 1: make uniform the beginning of the sentences to capital letter or not, but some of the symptoms and treatment start with capital letter and others no.

We thank the reviewer for this note and changed the first letter in the table to a uniform beginning.

Reviewer 2 Report

This is a useful review of a complex and increasingly important subject. It needs a thorough re-reading to correct errors in English, such as line 15 in Abstract-weigh more?; extra comma in line 24; repeated error notices throughout the paper; incomplete sentence line 70-71; use of trail instead of trial in Therapy section and others. Also, should describe the figures at appropriate places in text. Conclusion says setmelanocortin is a treatment but text uses setmelanotide, Finally I suggest not directly recommending setmelanotide as treatment before bariatric surgery but merely considering it because there are several side effects. Also I would state clearly that the authors have no financial interest in this treatment agent, or if they do, state this.

Author Response

First, we thank the reviewer for the generally positive evaluation of our review article. We are also grateful to the reviewer for the important points, which we fully addressed in the revised manuscript.

  • This is a useful review of a complex and increasingly important subject. It needs a thorough re-reading to correct errors in English, such as line 15 in Abstract-weigh more?; extra comma in line 24; repeated error notices throughout the paper; incomplete sentence line 70-71; use of trail instead of trial in Therapy section and others. Also, should describe the figures at appropriate places in text. Conclusion says setmelanocortin is a treatment but text uses setmelanotide,

We thank the reviewer for this important comment. We are sorry for the errors and corrected them. The missing links to the figures occurred with an error message due to technical problems with Microsoft Word and the conversion into a pdf file. We fixed this and references are now visible. 

  • Finally, I suggest not directly recommending setmelanotide as treatment before bariatric surgery but merely considering it because there are several side effects.

We are thankful for the important feedback. According to recommendation of the reviewer, we included a sentence about the side effects as well as a part with other therapy options:

Other therapy options are controlled and restrictive diets, physical activity pro-grams, psychomotricity and hormone substitution including the familial environment to increase the efficacy of the treatment [71,5]. The success of a bariatric surgery de-pends on the method as well as the familial environment, while vertical gastroplasty seems to be the best variant for monogenetic obesity according to Huvenne et al. [5].

Setmelanotide should be considered as a therapy option taking into account the side effects.

  • Also, I would state clearly that the authors have no financial interest in this treatment agent, or if they do, state this.

We thank the reviewer for this advice. We added the sentence, that no financial interest in the treatment agent exists.

The authors have no financial interest.

Reviewer 3 Report

Several case series of extreme early-onset obesity due to mutations in the human leptin receptor (LEPR) gene have been reported.

Authors to focus on compound heterozygous mutations of the leptin receptor and their effects on health as well as the possible therapy options in humans and animal models, in this work.

  1. I recommend:
    1. Clarification of the study design (narrative review, scope review, etc.) and explanation of the literature selection protocol
    2. Because, there is a disturbing imbalance between parts of the manuscript, expand on part 2.4 (Function) and 5. (Animal models of compound heterozygous Lepr mutations)
    3. Describe the roles of single nucleotide polymorphisms SNAP
    4. Explain the insertions: (Error! Reference source not found.)
    5. Many typos in part References (should be adapted to the requirements of the journal)

Author Response

First, we thank the reviewer for the generally positive evaluation of our review article. We are also grateful to the reviewer for the important points, which we fully addressed in the revised manuscript.

Several case series of extreme early-onset obesity due to mutations in the human leptin receptor (LEPR) gene have been reported. Authors to focus on compound heterozygous mutations of the leptin receptor and their effects on health as well as the possible therapy options in humans and animal models, in this work.

I recommend:

  • Clarification of the study design (narrative review, scope review, etc.) and explanation of the literature selection protocol

We thank the reviewer for this comment. As study design we used a systematic review.

The literature about compound heterozygous leptin receptor mutations is limited and all papers addressing this theme are reviewed in the article in an objective matter. Articles about the theoretical background (introduction, structure, signaling, function) are found by searching at https://pubmed.ncbi.nlm.nih.gov/ while original articles were preferred.

  • Because there is a disturbing imbalance between parts of the manuscript, expand on part 2.4 (Function) and 5. (Animal models of compound heterozygous Lepr mutations)

We are thankful for the important comment and expended part 2.4 function with the following:

Women suffering from mutations in leptin or LEPR could be infertile regarding to low levels of follicle-stimulating and luteinizing hormone leading to missing attain of pubertal maturity and hypogonadotropic hypogonadism [52,48]. On the other hand, it is reported, that men with leptin receptor mutations getting fertile with the beginning of puberty [53]. Fertility depends on the location of the mutation, while STAT3 and 5 signaling do not influence fertility, the exact mechanism is not known jet [41,54]. Also, growth is not depending on STAT3 signaling of LEPR, as studies with STAT3 in LEPR deficient mice exhibited increased growth while whole brain knock out animals demonstrated reduced body length [41,55]. Reports on growth in humans with im-paired LEPR function are very heterogeneous. Decreased growth hormone response and insulin-like growth factor 1 (IGF1) concentrations or Leptin as counterpart are still discussed.

Leptin signaling is also able to regulate glucose homeostasis as shown in ob/ob mice [56]. Thereby, the signaling is taking place in hypothalamus as well as within the PI3K part of the LEPR signaling pathway [41,57,58].

The nutritional status and concentrations of leptin within serum modulate the T-cell immune function [59]. Thereby, low leptin levels lead to a reduced CD4+ T-cell number, a decreased T-cell proliferation and a higher compensatory B-cell number leading to an altered cytokine release after nonspecific and specific stimuli [60,48].

For part 5 is just one animal model described, which is not published until now. This makes it difficult to expand this part.

  • Describe the roles of single nucleotide polymorphisms SNP

We thank the reviewer for the feedback and added a part describing SNP:

Other SNPs within the LEPR are associated with a higher BMI in a Korean or Spanish population [64,65], insulin resistance or adiponectin serum level [64] as well as obesity and leptin serum level [66]. Thereby, the analysis of SNPs helps to identify dis-ease-causing variants in a gene within a population [67].

  • Explain the insertions: (Error! Reference source not found.)

We thank for this important comment. The references appeared with an error message due to technical problems with the Microsoft Word document converting into a pdf file. We fixed this and the references are now visible.

  • Many typos in part References (should be adapted to the requirements of the journal)

We thank the reviewer for this point. We checked all references and corrected them according to the requirements of the journal.

Round 2

Reviewer 2 Report

The revised version answers my concerns adequately.

Reviewer 3 Report

Paper was improved, according to reviewer's suggestion

This manuscript is a resubmission of an earlier submission. The following is a list of the peer review reports and author responses from that submission.